# Bioenergetic control of soil carbon dynamics across depth

**Ludovic Henneron** [1,2] ✉, **Jerôme Balesdent**[3,9], **Gaël Alvarez** [1], **Pierre Barré**[4], **François Baudin** [5], **Isabelle Basile-Doelsch**[3], **Lauric Cécillon** [2,4], **Alejandro Fernandez-Martinez** [6], **Christine Hatté** [7,8] **& Sébastien Fontaine**[1]

Soil carbon dynamics is strongly controlled by depth globally, with increasingly slow dynamics found at depth. The mechanistic basis remains however controversial, limiting our ability to predict carbon cycle-climate feedbacks. Here we combine radiocarbon and thermal analyses with long-term incubations in absence/presence of continuously $^{13}C/^{14}C$-labelled plants to show that bioenergetic constraints of decomposers consistently drive the depth-dependency of soil carbon dynamics over a range of mineral reactivity contexts. The slow dynamics of subsoil carbon is tightly related to both its low energy density and high activation energy of decomposition, leading to an unfavourable 'return-on-energy-investment' for decomposers. We also observe strong acceleration of millennia-old subsoil carbon decomposition induced by roots ('rhizosphere priming'), showing that sufficient supply of energy by roots is able to alleviate the strong energy limitation of decomposition. These findings demonstrate that subsoil carbon persistence results from its poor energy quality together with the lack of energy supply by roots due to their low density at depth.

Earth's soils store more carbon in soil organic matter (SOM) than the global vegetation and atmosphere combined[1]. The fate of this large terrestrial reservoir of carbon under global change remains however a major uncertainty limiting our ability to accurately predict carbon cycle-climate feedbacks[2,3]. While most of our research effort has so far focused on soil organic carbon (SOC) located up to 30 cm deep (topsoil), a major portion corresponding to around half of the global SOC stock is stored in deeper soil layers (subsoil)[1,4]. Deep SOC formation can derive from dissolved organic matter and colloidal organo-mineral particles percolating downward through the soil profile, as well as organic matter inputs by deep plant roots, accumulation of eroded soil downhill and bioturbation by soil fauna such as earthworms[4–7]. Depth is the primary driver of SOC dynamics globally, with SOC turnover time

increasing with depth from decades or centuries in topsoil to millennia in subsoil[8,9]. There is still much controversy about the underlying mechanisms explaining the slow dynamics of deep SOC[5,10]. Unravelling the mechanistic basis of the depth-dependency of SOC dynamics is critical not only for improving our fundamental knowledge about SOC dynamics, but also to enhance our ability to forecast the response of the large reservoir of SOC to global change.

The high persistence of deep SOC has been largely attributed to stabilisation mechanisms reducing its accessibility to microbial decomposers and their extracellular enzymes[11]: either by mineral protection related to its large proportion associated with reactive minerals[7,12–14], or by physical separation related to its sparse density and associated high spatial heterogeneity in the soil matrix[15–17].

[1]INRAE, VetAgro Sup, Université Clermont Auvergne, UMR Ecosystème Prairial, Clermont-Ferrand, France. [2]Normandie Université, UNIROUEN, INRAE, ECODIV, Rouen, France. [3]Aix Marseille Univ, CNRS, IRD, INRAE, CEREGE, Aix en Provence, France. [4]Ecole normale supérieure, CNRS, IPSL, Université PSL, Laboratoire de Géologie, Paris, France. [5]CNRS, Sorbonne Université, ISTeP, Paris, France. [6]Université Grenoble Alpes, Université Savoie Mont Blanc, CNRS, IRD, IFSTTAR, ISTerre, Grenoble, France. [7]CEA, CNRS, UVSQ, Université Paris-Saclay, Laboratoire des Sciences du Climat et de l'Environnement, Gif-sur-Yvette, France. [8]CSE, Silesian University of Technology, Institute of Physics, Gliwice, Poland. [9]Deceased: Jerôme Balesdent. ✉e-mail: ludovic.henneron1@univ-rouen.fr

Suboptimal environmental conditions such as low temperature and anaerobic conditions have additionally been mentioned[10,11], and obviously represent key drivers of deep SOC persistence for permafrost and peatlands. Empirical evidence supporting their importance for mineral well-aerated soils remains however limited[12,14,16]. Because energy limitation of microbial metabolism is a fundamental property of deep environments on Earth[18], it has also been proposed that the high persistence of deep SOC could be related to a strong energy limitation of decomposers at depth[17,19,20].

Some theoretical models suggest that microbes decomposing persistent SOC are mostly limited by the metabolic energy needed for the biosynthesis of exoenzymes catalysing SOC depolymerisation and solubilisation, which are the major bottlenecks restricting microbial uptake and respiration of SOC[21,22]. Metabolising SOC must indeed yield greater energy to decomposers than they invest in exoenzyme production to acquire this energy. If the resulting 'return-on-energy-investment' is negative or does not reach maintenance energy needs, decomposers then starve and their growth and exoenzyme production eventually stop, thereby allowing SOC to persist[23,24]. Consistent with this idea, a bioenergetic framework has been proposed to assess the energy quality of SOC related to its 'return-on-energy-investment' for decomposers[25–28]. Deep SOC has previously been shown to feature a

smaller energy density[29] and larger thermal stability[30] than surface SOC, suggesting a decline in SOC energy quality with depth.

Energy limitation of deep SOC decomposition could also provide a mechanistic underpinning to the large destabilization of deep SOC associated with fresh organic matter supply[20]. Plant living roots have for instance been shown to induce an acceleration of deep SOC decomposition[17], corresponding to a 'rhizosphere priming' phenomenon[31]. Rhizodeposition indeed represents a major source of fresh bioavailable and energy-rich substrates for soil decomposers[32,33]. Rhizodeposition could thereby trigger the decomposition of persistent SOC having poor energy quality via microbial co-metabolism because it can subsidise the energy cost of exoenzyme production[21,22]. Rhizosphere priming could also be driven by rhizosphere processes increasing SOC accessibility such as root exudation of organic ligands disrupting organo-mineral associations[34–36], or root uptake of water and nutrients intensifying drying-rewetting cycles and breakage of soil aggregates[31]. The lack of supply of energy in different forms, either metabolic, chemical or mechanical, by plant roots due to their low density at depth could thus be at play in explaining deep SOC persistence[4,20].

Direct experimental evidence linking deep SOC dynamics to both its bioenergetic signature and sensitivity to rhizosphere priming is still lacking to date. We thus investigate here the bioenergetic control of the depth-dependency of SOC dynamics in temperate well-aerated mineral soils. Specifically, we hypothesise that the slow dynamics of deep SOC is related to its poor energy quality together with the lack of energy supply by plant roots in subsoil. To test this hypothesis, we combine a characterisation of soil biogeochemical properties, including radiocarbon and thermal analyses, with a complementary experiment involving long-term soil incubations under controlled conditions in presence or absence of continuously $^{13}C/^{14}C$-labelled plants. To assess the robustness of our hypothesis according to soil mineral reactivity context, we replicate our study in three soil profiles with contrasting mineralogy: a cambisol, a vertisol and an andosol which are respectively characterised by moderate, high and very high mineral reactivity and deep SOC persistence based on a priori knowledge[14]. Our results demonstrate that the slowing of SOC dynamics with depth is consistently linked to a degradation of SOC energy quality, that is a smaller energy density and larger activation energy of decomposition in connection with its chemistry and interactions with soil minerals, together with a decline in the density of roots and their energy supply. These findings provide insights into the bioenergetic control of SOC persistence and indicate that an increase in plant rooting depth induced by global change could threaten the storage of millennia-old SOC in deep layers.

## Results

### Bioenergetic control of SOC dynamics across depth

Our biogeochemical characterisation of soil showed a clear in situ differentiation of SOM properties across depth that was consistently found for each of the three soil profiles studied despite their contrasting mineralogy (Fig. 1, RC1). Relative to topsoil, the subsoil was characterised by smaller concentration ([SOC]) and radiocarbon signature ($\Delta^{14}C$) of SOC, as well as lesser proportion of particulate organic matter (fPOM, Table 1). Subsoil was in contrast characterised by higher $\delta^{13}C$ signature of SOC indicative of greater microbial processing[37], as well as greater proportion of mineral-associated organic matter (fMAOM). Microbial biomass per unit soil mass and SOC were both smaller for subsoil than topsoil. A modelling approach was used to estimate SOC dynamics from radiocarbon ($\Delta^{14}C$) measurements of SOC and we found a large increase in SOC turnover time with depth, increasing on average 6.9-fold from 1096 to 7567 years.

We adopted a bioenergetic framework to assess how SOC dynamics could be related to its energy quality for microbial decomposers[25–28], defined here as the net energy gain that

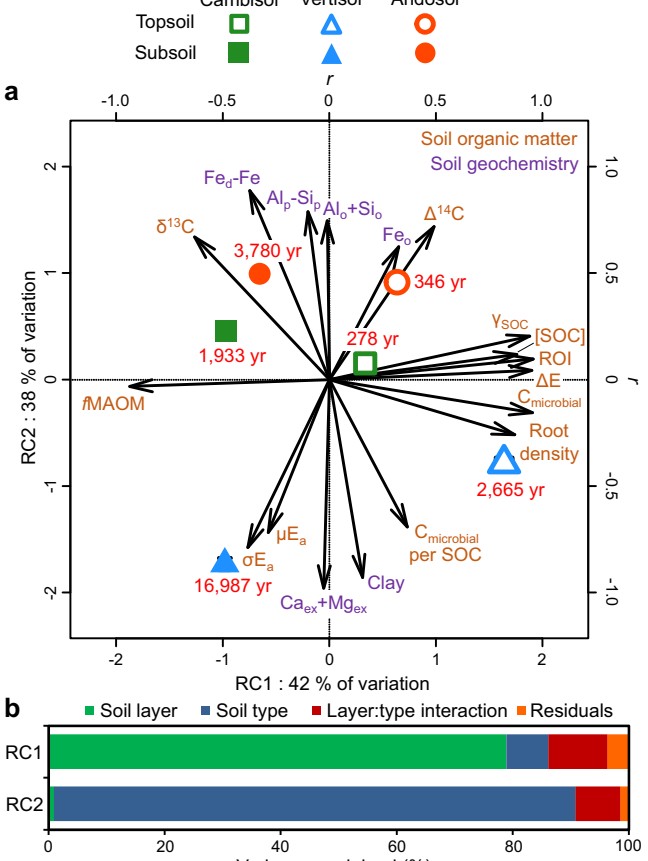

**Fig. 1 | Variation in soil biogeochemical properties among soil types and layers. a** ordination by rotated principal component analysis (n = 24 soil cores). Arrows indicate correlations between soil properties and rotated component (RC) axes. Large symbols with error bars show treatment means ± standard errors. The numbers in red indicate the turnover time of soil organic carbon (SOC) based on radiocarbon signature for each treatment. The subscripts $_{ex}$, $_d$, $_o$, and $_p$ indicate exchangeable, dithionite, oxalate, and pyrophosphate extracts, respectively. See Table 1 for other abbreviations. **b** variance partitioning of RC axis values across experimental factors.

**Table 1 | Depth variation in soil organic carbon (SOC) properties averaged across the three soil types**

| | Topsoil | Subsoil | Depth effect size | Partial η² |
|---|---|---|---|---|
| SOC concentration ([SOC], g SOC kg⁻¹ soil) | $85.5 \pm 5.6$ | $34.7 \pm 5.6$ | $-59.4\%$[a] | 0.57 |
| fPOM (% SOC) | $8.3 \pm 0.6$ | $2.7 \pm 0.3$ | $-67.9\%$[a] | 0.82 |
| fMAOM (% SOC) | $91.7 \pm 0.6$ | $97.3 \pm 0.3$ | $+6.1\%$[a] | 0.82 |
| $\delta^{13}C$ (‰) | $-26.85 \pm 0.12$ | $-25.88 \pm 0.27$ | $+0.98$[b] | 0.80 |
| $\Delta^{14}C$ (‰) | $-75.8 \pm 34.2$ | $-375.8 \pm 61.1$ | $-300$[b] | 0.88 |
| SOC turnover time (τ, year) | $1096 \pm 340$ | $7566 \pm 2033$ | $+590\%$[a] | 0.56 |
| Energy density ($\Delta E$, kJ mol⁻¹ SOM) | $175.1 \pm 4.5$ | $126.7 \pm 4.5$ | $-27.6\%$[a] | 0.85 |
| Degree of reduction of SOC ($\gamma_{SOC}$) | $3.05 \pm 0.06$ | $2.26 \pm 0.08$ | $-26.0\%$[a] | 0.83 |
| Hydrogen Index (HI, g $H_xC_y$ kg⁻¹ SOC) | $229.1 \pm 16.9$ | $114.5 \pm 10.6$ | $-50.0\%$[a] | 0.68 |
| Oxygen Index (OI, g $O_2$ kg⁻¹ SOC) | $180.1 \pm 9.6$ | $222.2 \pm 19.1$ | $+23.4\%$[a] | 0.55 |
| Mean activation energy ($\mu E_a$, kJ mol⁻¹ SOM) | $159.5 \pm 0.4$ | $162.0 \pm 0.9$ | $+2.4$[b] | 0.61 |
| SD of activation energy ($\sigma E_a$, kJ mol⁻¹ SOM) | $16.10 \pm 0.06$ | $16.80 \pm 0.28$ | $+0.70$[b] | 0.43 |
| $T_{90}$-$H_xC_y$-pyrolysis (°C) | $521.3 \pm 0.7$ | $532.8 \pm 0.9$ | $+11.5$[b] | 0.82 |
| $T_{50}$-$CO_2$-pyrolysis (°C) | $387.2 \pm 0.9$ | $394.3 \pm 2.6$ | $+7.1$[b] | 0.51 |
| $T_{50}$-$CO_2$-oxidation (°C) | $422.5 \pm 2.1$ | $437.8 \pm 5.2$ | $+15.3$[b] | 0.56 |
| Return-on-energy-investment (ROI, $\Delta E$:$\mu E_a$) | $1.10 \pm 0.03$ | $0.78 \pm 0.03$ | $-28.7\%$[a] | 0.85 |
| Microbial biomass ($C_{microbial}$, g C kg⁻¹ soil) | $1.25 \pm 0.17$ | $0.32 \pm 0.03$ | $-74.1\%$[a] | 0.73 |
| Microbial biomass per unit SOC (g C kg⁻¹ SOC) | $15.2 \pm 0.9$ | $10.9 \pm 1.4$ | $-28.1\%$[a] | 0.37 |
| Root density (g dm⁻³) | $3.21 \pm 0.82$ | $0.14 \pm 0.03$ | $-95.6\%$[a] | 0.52 |

fPOM fraction of particulate organic matter; fMAOM, fraction of mineral-associated organic matter; $\mu E_a$ and $\sigma E_a$, mean and standard deviation of the activation energy of soil organic carbon (SOC) decomposition; Return-on-energy-investment (ROI) is the ratio of $\Delta E$ to $\mu E_a$; $T_{90}$-$H_xC_y$-pyrolysis, temperature at which 90 % of $H_xC_y$ was evolved during pyrolysis; $T_{50}$-$CO_2$-pyrolysis and $T_{50}$-$CO_2$-oxidation, temperatures at which 50 % of $CO_2$ was evolved during pyrolysis and oxidation, respectively. Mean ± standard errors (n = 12 soil cores, including 3 soil types with 4 replicates each).
[a]Relative change in soil properties for subsoil relative to topsoil: this effect size was computed as $(\bar{x}_{subsoil} - \bar{x}_{topsoil})/\bar{x}_{topsoil}$.
[b]Absolute change in soil properties for subsoil relative to topsoil: this effect size was computed as $\bar{x}_{subsoil} - \bar{x}_{topsoil}$. Partial η² indicates the proportion of variance associated with the depth effect after accounting for soil type effect. See Supplementary Table 1 for depth variation in soil properties for each soil type, and Supplementary Table 2 for statistical results.

decomposers could realise for a given energy investment in SOC acquisition. The energy quality of SOC was quantified using the return-on-energy-investment (ROI) parameter[26], which is the ratio between (i) the energy density of SOC ($\Delta E$, energy content per unit SOC) representing the amount of energy that decomposers could gain by SOC catabolism[38], and (ii) the mean activation energy ($\mu E_a$) of SOC decomposition representing the energetic barriers to its acquisition by decomposers[39]. The energy density of SOC and the activation energy of its decomposition were respectively measured by differential scanning calorimetry and evolved gas analysis during ramped combustion[25,28].

This approach allowed us to reveal an important shift in SOC bioenergetic signature with depth (Fig. 1, Table 1, Supplementary Fig. 1). The energy density of SOC ($\Delta E$) was indeed 27.6% smaller for subsoil than topsoil (decline from 175.1 to 126.7 kJ mol⁻¹ SOM with depth). The degree of reduction of SOC ($\gamma_{SOC}$) was accordingly also lower at depth (decline from 3.05 to 2.26 with depth), consistent with the smaller hydrogen content (HI) and larger oxygen content (OI) of SOM. In contrast, the mean and standard deviation of the activation energy of SOC thermal decomposition were greater for subsoil than topsoil ($\mu E_a$, +2.4 kJ mol⁻¹ SOM; $\sigma E_a$, +0.70 kJ mol⁻¹ SOM), consistent with the larger thermal stability of deep SOC (higher $T_{90}$-$H_xC_y$-pyrolysis, $T_{50}$-$CO_2$-pyrolysis and $T_{50}$-$CO_2$-oxidation). Accordingly, the return-on-energy-investment parameter (ROI) declined with depth from 1.10 to 0.78, which was strongly related to the increase in SOC turnover time with depth (Fig. 2a). We also observed a very strong decline in root density with depth from 3.21 to 0.14 g dm⁻³ (Table 1), which was tightly related to the increase in SOC turnover time with depth as well (Fig. 2b).

### Root effects on SOC decomposition across depth
In order to investigate the consequences of the variation in root density across depth for SOC decomposition, we performed a complementary experiment involving long-term soil incubations in presence or absence of continuously ¹³C/¹⁴C-labelled plants. This dual-labelling allowed us to simultaneously partition soil and plant sources into C fluxes and pools, and determine the mean age of native SOC respired[17,20]. In an effort to decouple the natural covariation in the vertical distribution of root density and SOC properties[4], we grew a deep-rooting grass species (Dactylis glomerata) in soil columns made of intact soil cores derived exclusively from either topsoil or subsoil (Supplementary Fig. 2). As SOC dynamics can be very sensitive to physical disturbance, especially for subsoil[16,17], we took care to preserve the soil structure as undisturbed as possible during the sampling and incubations. We performed two series of incubations under moisture and temperature-controlled conditions. The first series of incubations performed throughout plant growth consisted of regularly measuring the respiration of the plant-soil system across time (Fig. 3a). The second series of incubations performed at the end of the experiment consisted of retrieving the original soil cores and measuring the respiration of the root-soil system at different depths in the soil column and thus at different root densities (Fig. 4a). The density of roots indeed declined from 3.57 to 0.51 g dm⁻³ with depth in the soil column, these values being close to the decline in root density with depth observed in the field.

For both incubation series, we found that the native SOC decomposition rate ($k_{SOC}$, respiration of pre-existing SOC per unit SOC stock) in the absence of roots was on average 3.0-fold smaller for subsoil relative to topsoil across the three soil types (Figs. 3b, 4b), consistent with the slower SOC dynamics found for subsoil than topsoil based on radiocarbon measurements. Native SOC decomposition included here the respiration of pre-existing SOM but also root litter. However, the amount of OC in pre-existing root litter represented on average only 2.6 and 0.3% of the amount of native SOC respectively for the topsoil and subsoil at the beginning of the experiment (Supplementary Table 6). By the end of the experiment, 90.1% of pre-existing root masses were lost on average, and pre-existing dead roots OC represented on average 0.27% of the amount of native SOC for the

topsoil. This indicates that our estimation of $k_{SOC}$ largely reflects the decomposition of native SOM, rather than pre-existing root litter.

The radiocarbon signature of respired $CO_2$ in unplanted subsoil cores at the end of the experiment revealed that the mean age of native SOC respired by decomposers was within the range of initial SOC dominated by millennia-old pools, that were around 1950 and 2900 years respectively for the cambisol and andosol (Fig. 5). This result indicates that the small amounts of young and labile organic matter present in subsoil (root litter and particulate organic matter, Table 1) were probably largely exhausted after nearly 1 year of incubation, leading microbes to decompose old SOC pools[17].

The presence of roots induced an acceleration of $k_{SOC}$ corresponding to a rhizosphere priming effect (RPE) for both topsoil and subsoil (Figs. 3b, c, 4b, c). In the first incubation series, we observed an increase in $k_{SOC}$ that was associated with an increase in the respiration of plant-derived OC related to plant growth over time (Fig. 3a). In the second incubation series, we observed a concomitant increase in $k_{SOC}$, living root density, respiration of root-derived OC and net rhizodeposition with decreasing soil column depth (Fig. 4a). Consequently, we found strong positive relationships of $k_{SOC}$ with respiration of plant-derived OC in the first incubation series (Fig. 3b), as well as with living root density, respiration of root-derived OC and net rhizodeposition in the second incubation series (Fig. 4b, d, e).

For both incubation series, we observed that the $k_{SOC}$ of topsoil and subsoil converged toward high levels at high plant activity or root density (Figs. 3b, 4b), so that the acceleration of $k_{SOC}$ induced by roots (RPE) was proportionally much higher for subsoil than topsoil (Figs. 3c, 4c). In the first incubation series, the average RPEs across soil types were for instance +64% for topsoil and +411% for subsoil at a given high level of respiration of plant-derived OC (9.94 g C–CO$_2$ m$^{-2}$ day$^{-1}$), corresponding to a 6.4-fold difference (Fig. 3c). In the second incubation series, we similarly found that the average RPEs across soil types were +138% for topsoil and +535% for subsoil at a given high level of living root density (3.57 g dm$^{-3}$), corresponding to a 3.9-fold difference (Fig. 4c). We also observed a strong asymptotic increase toward very high mean age of native deep SOC respired by decomposers with increasing living root density at the end of the experiment, reaching

around 9,000 and 17,000 years respectively for the cambisol and andosol at high root density (Fig. 5).

Overall, we found that SOC bioenergetic signature properties and root density were as good or better predictors of SOC dynamics parameters than more classical predictors of SOC stability such as $f$POM *versus* $f$MOAM, $\delta^{13}C$ or [SOC] (Supplementary Figs. 3, 4).

## Mineral control of SOC dynamics across depth

The field characterisation of soil also confirmed strong differences in mineralogy among soil types (Fig. 1, RC2). The vertisol and andosol contained high amounts of reactive secondary minerals of different mineral composition. The andosol was characterised by high concentrations of organically complexed metals (Al$_p$-Si$_p$), as well as short-range-ordered (SRO) metal oxyhydroxides (Fe$_o$, e.g. ferrihydrite) and aluminosilicates (Al$_o$ and Si$_o$, e.g. allophane) forming covalent bonds with SOC through ligand exchange. The vertisol was characterized by high concentrations of phyllosilicate minerals (clay) such as smectite and halloysite based on XRD analyses (Supplementary Table 1), as well as divalent metal cations (Ca$_{ex}$ + Mg$_{ex}$) involved in organo-mineral associations through cation bridging. The cambisol was characterized by smaller concentrations of reactive minerals, mostly composed of Fe oxides in both crystalline (Fe$_d$–Fe$_o$) and SRO oxyhydroxide (Fe$_o$) forms as well as divalent metal cations along with kaolinite and vermiculite clay minerals. We concomitantly observed higher turnover time of subsoil SOC for the andosol (3780 years) and vertisol (16,987 years) than the cambisol (1933 years, Fig. 1).

The vertisol SOC also showed high thermal stability (T$_{50}$-CO$_2$-pyrolysis, T$_{50}$-CO$_2$-oxidation), low radiocarbon signature ($\Delta^{14}$C) and oxygen content (OI) for both the topsoil and subsoil. This suggests a substantial abundance of pyrogenic SOC that could partly explained the unusually high turnover time of vertisol SOC[40], though a partial contribution of fossil C from volcanic $CO_2$ emission or inherited from the parent material could also possibly be involved[7].

Overall, the smaller $k_{SOC}$ in the absence of roots and larger RPE for subsoil than topsoil was a pattern that has been consistently found for each of the three soil types with contrasting mineralogy studied here (Figs. 3c, 4c). However, we observed important differences in

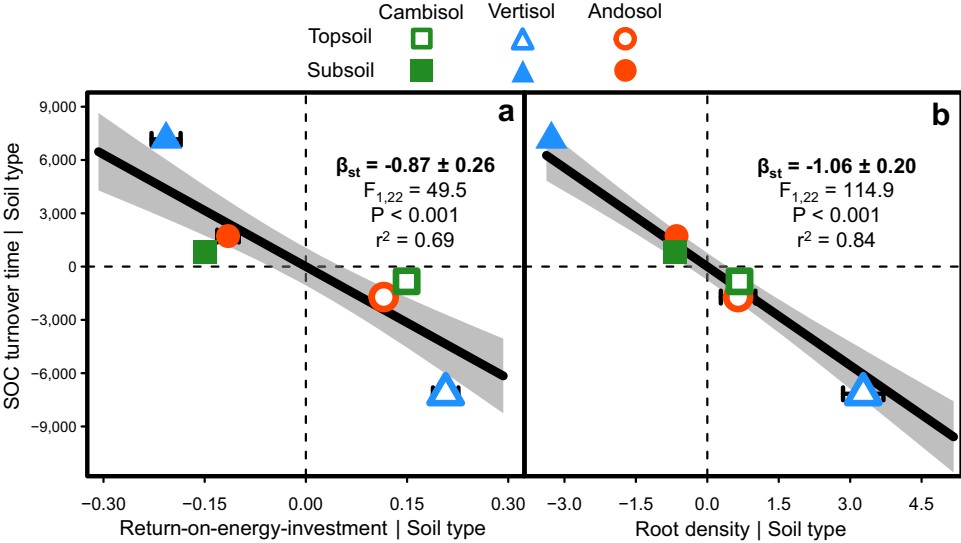

**Fig. 2 | Energy quality and root density as drivers of the depth-dependency of soil carbon dynamics.** Partial regression plots relating the depth variation in the turnover time of soil organic carbon (SOC) with those of the return-on-energy-investment parameter (**a**) and root density (**b**, n = 24 soil cores). The residuals of linear models fitting each variable with 'soil type' as a fixed effect were used here so that each variable was normalised for variation across soil types. The return-on-energy-investment parameter was calculated as the ratio between the energy density of SOC and its mean activation energy as determined by thermal analyses, and was used here as an indicator of SOC energy quality. Symbols with error bars show treatment means ± standard errors (n = 4 replicate soil cores). Polygons around regression lines represent 95% confidence intervals. β$_{st}$ are regression slope coefficients ± 95% confidence intervals standardised by range. Significant β$_{st}$ are in bold (two-sided F-test).

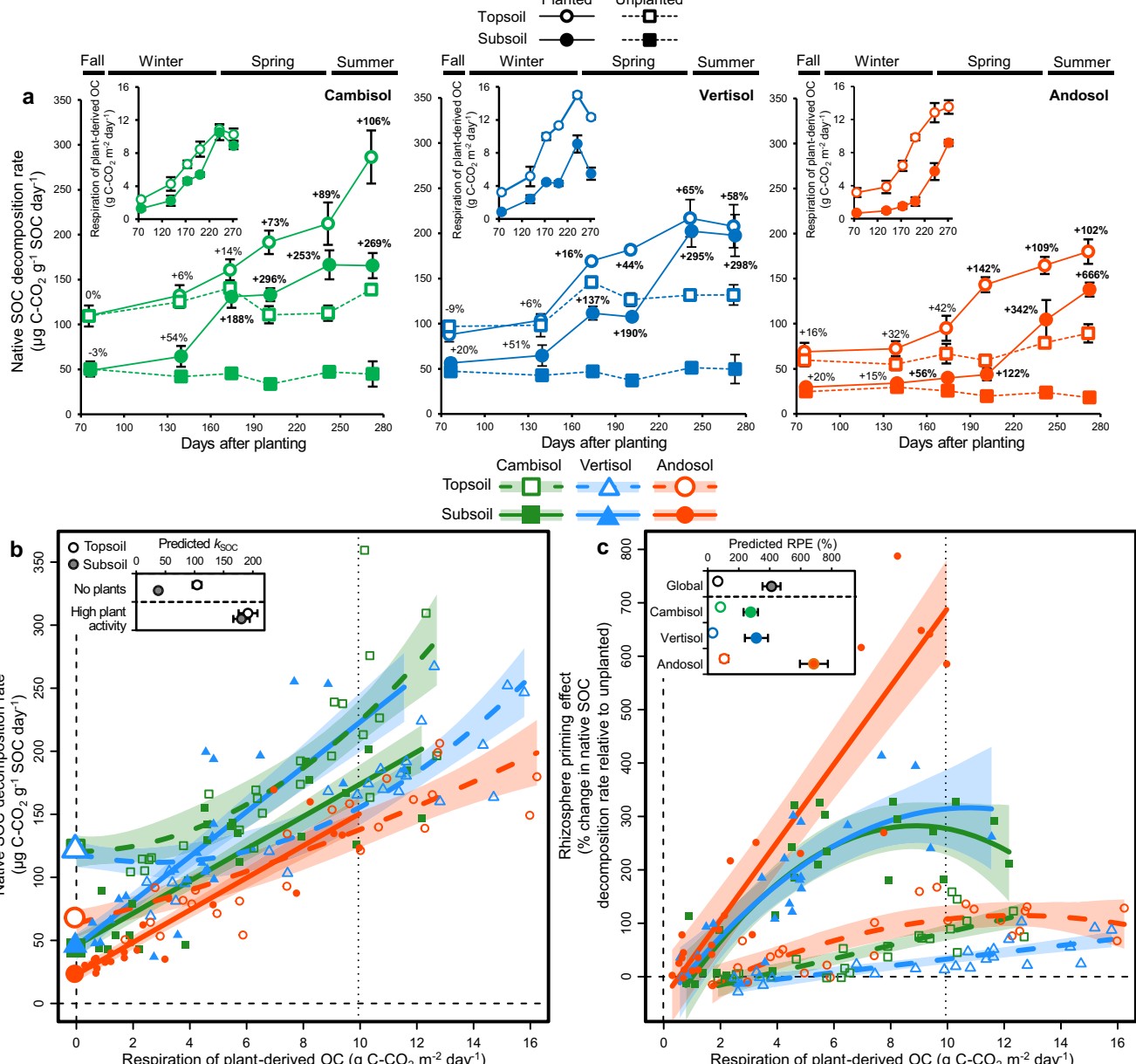

**Fig. 3 | Root effects on the decomposition of native soil organic carbon (SOC) across time. a** Temporal variation in native SOC decomposition rate ($k_{SOC}$) and respiration of plant-derived organic carbon (OC, insets) at the microcosm scale in the first series of incubations ($n = 288$ microcosm incubations). The respiration of plant-derived OC includes both plant autotrophic respiration and soil microbial heterotrophic respiration derived from rhizodeposits. Symbols with error bars show treatment means ± standard errors ($n = 4$ replicate microcosms). Numbers above the symbols represent rhizosphere priming effects (RPEs, in % change in $k_{SOC}$ for planted soils relative to the unplanted control). Significant RPEs are in bold (two-sided t-test). **b, c** Response of native SOC decomposition rate (**b**, $k_{SOC}$, $n = 288$

microcosm incubations) and rhizosphere priming effect (**c**, RPE, $n = 144$ planted microcosm incubations) to the variation in respiration of plant-derived OC across time in the first incubation series. Symbols with error bars along the vertical dashed line in (**b**) show treatment means ± standard errors of unplanted controls ($n = 24$ microcosm incubations). Polygons around regression lines represent 95% confidence intervals. The inset in (**b**) show $k_{SOC}$ values averaged across soil types in the absence of roots and at high level of plant activity as predicted for a common high value of the predictor shown by the vertical dotted line. The insets in (**c**) show predicted RPE values across treatments at high level of plant activity. Statistical results are shown in Supplementary Table 3.

RPE among soil types for the subsoil. Indeed, the RPE at high plant activity was much higher for the andosol than the vertisol and cambisol in the first incubations series (Fig. 3c), while the RPE at high root density was higher for the andosol and to a lesser extent the vertisol than for the cambisol (Fig. 4c).

## Discussion
Our study combining radiocarbon and multiple thermal analyses methods along with plant isotopic labelling and long-term soil incubations provides a robust set of observations supporting the idea of a

bioenergetic control of the depth-dependency of SOC dynamics (Fig. 6). The high persistence of deep SOC was indeed related to its poor energy quality (Fig. 2a). More precisely, deep SOC decomposition required higher energy inputs to proceed (activation energy) while yielding less energy available for biosynthesis (energy density), leading to a less favorable 'return-on-energy-investment' for microbes decomposing this persistent SOC[25–28] (Table 1). These results provide evidence of an increasing energy limitation of SOC decomposition with depth. The slow dynamics of deep SOC was also strongly related to the low density of plant roots in subsoil (Fig. 2b). Our soil incubation

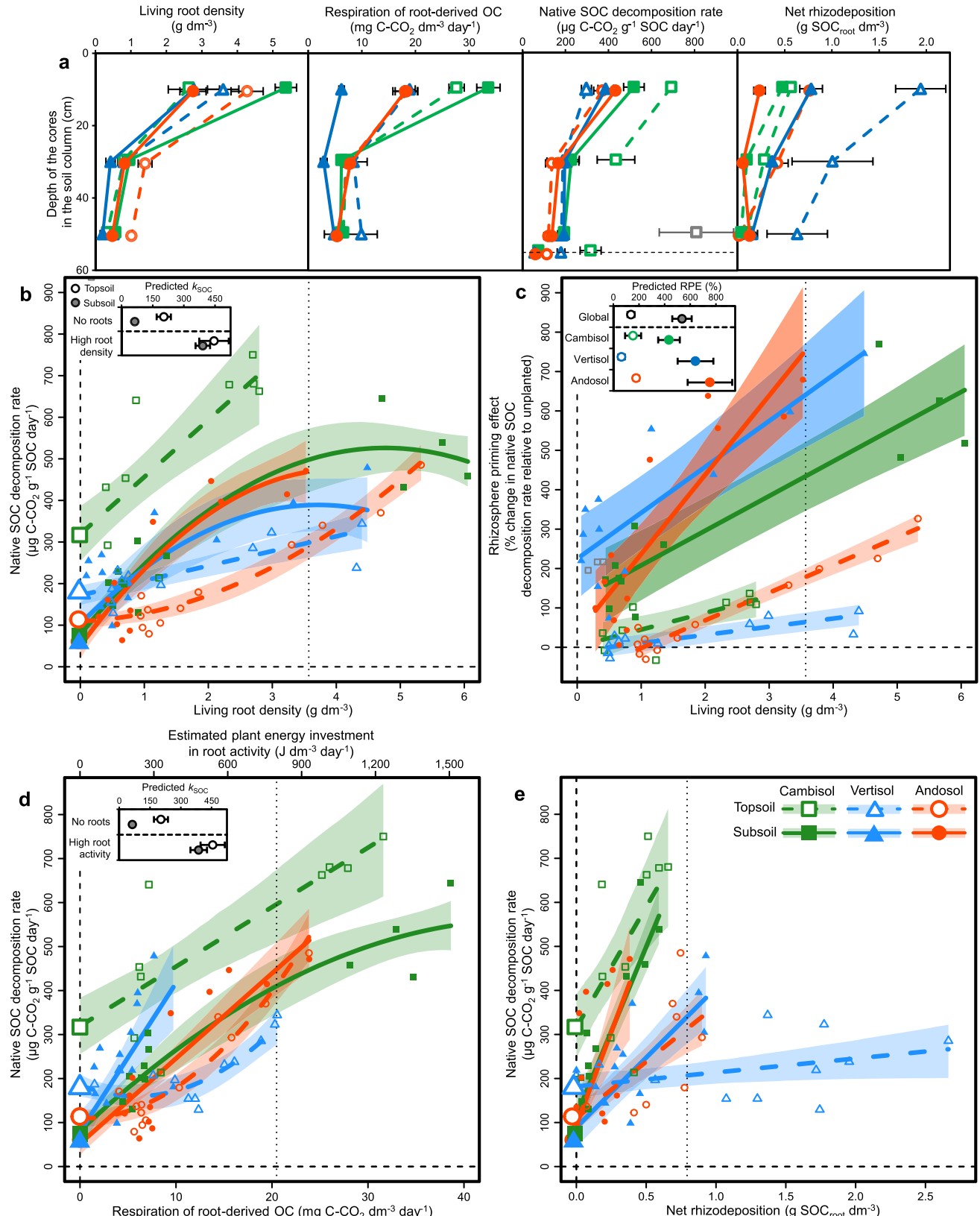

experiment showed that the decomposition rate of native SOC was much lower for subsoils than topsoils in the absence of roots, while subsoil and topsoil SOC decomposition converged toward greater levels at high root density typically found in topsoil (Figs. 3, 4). Such acceleration of deep SOC decomposition by root activity (rhizosphere priming) led to the respiration of millennia-old SOC (Fig. 5). These

results demonstrates that the persistent SOC found in subsoil could be destabilised by rhizosphere priming when root density and activity are high enough to provide the supply of energy able to alleviate the energy limitation of SOC decomposition[17,20]. This implies that the energy limitation of deep SOC decomposition related to its poor energy quality is exacerbated by the lack of energy supply by plant

**Fig. 4 | Root effects on the decomposition of native soil organic carbon (SOC) across soil column depth. a** Depth variation in living root density, respiration of root-derived organic carbon (OC), native SOC decomposition rate and net rhizodeposition at the soil core scale in the second incubation series 279 days after planting ($n = 144$ soil core incubations). Symbols with error bars show treatment means ± standard errors ($n = 4$ replicate soil cores). **b–e** Response of native SOC decomposition rate (**b**, **d**, **e**, $k_{SOC}$, $n = 144$ soil core incubations) and rhizosphere priming effect (**c**, RPE, $n = 72$ soil core incubations) to the variation in living root density (**b**, **c**), respiration of root-derived OC (**d**) and net rhizodeposition (**e**) across soil column depth in the second incubation series. The respiration of root-derived OC includes both root autotrophic respiration and soil microbial heterotrophic respiration derived from rhizodeposits, and was used here to estimate plant energy supply to soil by root activity (Methods section). Symbols with error bars along the horizontal dashed line in (**a**) and along the vertical dashed line in (**b**, **d**, **e**) show treatment means ± standard errors of unplanted controls ($n = 24$ soil core incubations). Polygons around regression lines represent 95% confidence intervals. Grey symbols represent outliers. Statistical results are shown in Supplementary Table 4.

roots due to their low density at depth, thus contributing of deep SOC persistence. These results also contribute to the growing body of evidence showing the greater sensitivity to priming of SOC with higher persistence because of a greater energy limitation of decomposition[41].

The shift in SOC bioenergetic signature with depth could be related to both its chemistry and interactions with soil minerals. Deep SOC had broader distributions of activation energy than surface SOC (Table 1), and this larger bond-strength diversity indicates larger formation of high-energy bonds with reactive minerals contributing to its persistence by mineral protection[39]. Accordingly, the smaller energy density of deep SOC could be partly ascribed to the greater proportion of MAOM requiring the breaking of organo-mineral bonds before decomposition, and conversely to the lesser proportion of energy-richer POM[42]. From a chemical point of view, deep SOC had smaller degree of reduction ($\gamma_{SOC}$, Table 1), which was related to its reduced content of highly energetic C−H bonds and often increased content of less energetic C−O bonds[25,43]. This is consistent with the idea of SOC becoming increasingly oxidised (less reduced) as decomposition proceed[44]. The $\gamma_{SOC}$ represents a measure of the chemical energy stored in organic matter, corresponding to the number of electron equivalents per SOC amount[38,43]. The $\gamma_{SOC}$ is widely known to control microbial carbon-use efficiency[38,45,46], that is the ratio of growth over C uptake. When $\gamma_{SOC}$ decreases below the degree of reduction of C in biomass of microbial decomposers ($\gamma \sim 4.2$), their biosynthesis become increasingly energy limited. We found here that $\gamma_{SOC}$ decreased with depth on average from 3.05 for topsoil to 2.26 for subsoil. This result highlights that the smaller energy density of deep SOC could also be ascribed to the fact that deep SOC is more oxidised[47].

Despite their contrasting mineralogy and deep SOC persistence, the three soil types studied showed a similar pattern of slower SOC dynamics with depth that was related to a decline in both SOC energy quality and root density (Fig. 2). This provides evidence that bioenergetic constraints of decomposers consistently drive the depth-dependency of SOC dynamics over a range of mineral reactivity contexts. It remained so far uncertain how mineral reactivity modulates the sensitivity of deep SOC to rhizosphere priming[35,36,48], despite soil mineralogy being largely recognised as a key control over deep SOC persistence[12,14]. We observed here important mineral control of the depth-dependency over root effects on SOC decomposition. In contrast with the idea that strong mineral protection of SOC provided by high mineral reactivity would lead to weak rhizosphere priming because of the low accessibility of SOC to microbial metabolism[48], our results interestingly showed that a greater mineral protection of deep SOC also involves a greater sensitivity to rhizosphere priming (Figs. 3c, 4c, 6). This was especially clear for the andosol rich in SRO minerals forming covalent bonds with SOC through ligand exchange[14,49]. This is consistent with recent studies showing that root exudation of organic acids acting as ligands can enhance SOC accessibility to decomposers by disruption of protective organo-mineral associations[36,50,51], thereby promoting greater rhizosphere priming for soil types with larger mineral reactivity[35].

Suboptimal environmental conditions have been mentioned as potentially important drivers of deep SOC persistence[10,11]. Our results largely confirmed empirical studies showing that their importance for mineral well-aerated soils remains limited[12,14,16]. The smaller decomposition rate of native SOC under temperature and moisture-controlled conditions for subsoil than topsoil in the absence of roots indeed demonstrates that the slowing of SOC dynamics with depth cannot be merely driven by abiotic conditions such as cold or water-logged pedoclimate and oxygen limitation at depth. Furthermore, our result of a decline in $\gamma_{SOC}$ with depth contrasts with the idea that increasingly anaerobic conditions within microaggregates with depth can lead to deep SOC persistence even for well-aerated soils[52]. This is because it should theoretically involve an increase in $\gamma_{SOC}$ by preferential preservation of highly reduced organic matter. This mechanism may however be central in permanently waterlogged soils in wetlands and peatlands[43,53].

The mechanisms controlling SOC dynamics are still actively debated. One of the leading paradigm of recent years postulates that SOC dynamics is largely controlled by stabilisation mechanisms reducing its accessibility to microbial decomposers and their extracellular enzymes[10,11,44]. Though energy flow through living systems has long been considered a major driver of biogeochemical cycling processes[54], there has been comparatively little interest until recently about energy limitation of SOC dynamics[55]. Our findings support the emerging view that SOC dynamics could also be controlled by bioenergetic constraints experienced by decomposers[20,25,56]. Bioenergetic constraints have often been viewed as a competing mechanism of SOC stabilisation relative to those restricting SOC accessibility such as physical separation and mineral protection[36,48]. We propose here that these mechanisms are not mutually exclusive. We even suggest that stabilisation mechanisms restricting SOC accessibility can often be interpreted based on bioenergetic constraints. For instance, physical separation of SOC from its decomposers implies large diffusion distance between them and enhanced probability of exoenzyme inactivation and decomposition product loss by diffusion away from decomposers or immobilisation by reactive minerals and cheater microbes[15,17,23]. For this reason, greater physical separation between SOC and its decomposers can lead to lower return-on-energy-investment in exoenzymes[57]. However, physical separation could often be overcome by mobility for flagellated bacteria or hyphal growth for fungi. Similarly, mineral protection restricts exoenzyme access to SOC, but microbial decomposers can produce organic ligands[34,50] or reactive oxygen species through oxidative (exo) enzymes and reactive metal intermediates[58,59]. These compounds are able to disrupt organo-mineral associations, thereby enhancing microbial access to previously mineral-protected SOC. Nevertheless, both of these microbial strategies to access SOC imply metabolic costs, so that reduced accessibility could lead SOC to persist ultimately because of the associated exacerbation of energy limitation for decomposers. Our study demonstrates that persistent SOC can still be decomposed by rhizosphere priming provided that plants invest energy to (i) subsidise these metabolic costs for decomposers through rhizodeposition of fresh energy-rich compounds, and/or (ii) directly increase SOC accessibility through rhizosphere processes leading to the breakdown of organo-mineral associations and aggregates. We therefore propose that the role of mechanisms controlling SOC accessibility for SOC dynamics cannot be fully understood without explicit consideration of the

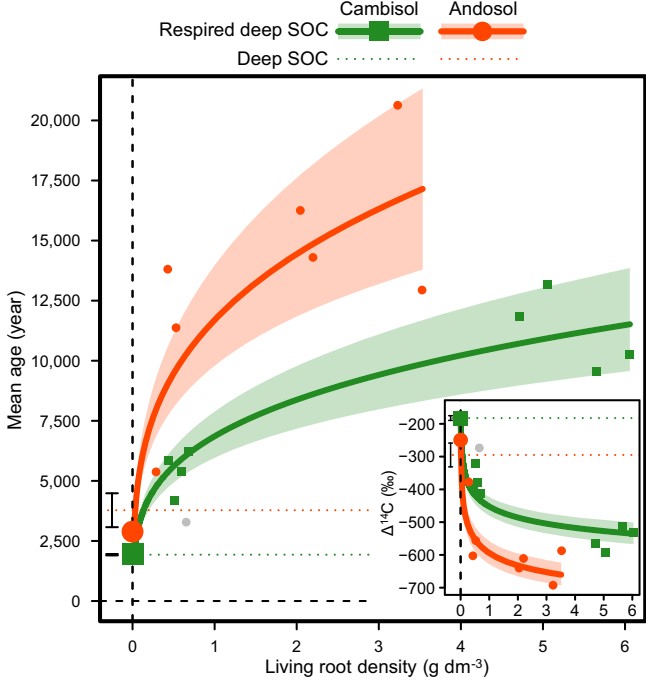

**Fig. 5 | Root effects on the mean age of subsoil organic carbon respired by decomposers.** Response of radiocarbon-based mean age of respired $CO_2$ derived from the decomposition of native deep soil organic carbon (SOC) to the variation in living root density across soil column depth at the soil core scale in the second incubation series 279 days after planting ($n = 24$ soil core incubations). The inset show radiocarbon signatures ($\Delta^{14}C$) of soil-respired $CO_2$ used to determine the mean age of respired native SOC. Since turnover time and mean age are identical under our model assumption of SOC being a homogeneous one-pool reservoir at steady-state, the mean ages were estimated using the same modelling approach used for turnover time as described in the methods section. Symbols with error bars along the vertical dashed line show treatment means ± standard errors of unplanted controls ($n = 4$ soil core incubations). Horizontal dotted lines with error bars show the means ± standard errors of mean ages (turnover times) and radiocarbon signatures of initial deep SOC. Polygons around regression lines represent 95% confidence intervals. The grey symbol is an outlier not included in regression model. Statistical results are shown in Supplementary Table 5.

bioenergetic constraints of decomposers and *vice versa*. Considering their interactive effects could thus prove valuable in our research effort towards developing a unified theory of biogeochemical controls on organic matter decomposition across Earth's environments[60].

There is growing evidence that global change drivers such as elevated $CO_2$[61,62] as well as climate warming and reduced precipitation[63,64] are increasing plant rooting depth in terrestrial ecosystems. The breeding and use of deep-rooting plant species resistant to drought has also been promoted as a land management adaptation to climate change in cropping systems[65]. Our results indicate that an increased root biomass allocation to deep soil layers previously featuring low root density can greatly enhance the decomposition of millennia-old SOC, which could threaten the storage of persistent SOC and potentially lead to a positive feedback on global warming[2]. This could however also promote SOC formation[31], but elevated $CO_2$ experiments focusing on topsoil SOC storage have shown that the loss of old SOC associated to rhizosphere priming usually largely offsets SOC formation[66,67], and can even lead to a decline in SOC storage when plant biomass is strongly stimulated[68]. Further research including a comprehensive SOC budget accounting for the balance between SOC formation and loss will therefore be crucial to evaluate the net effect of plant rooting depth on SOC storage in deep layers. Our results are at least

questioning the efficiency of using deep-rooting plant species as a SOC sequestration strategy[65,69].

## Methods
Further details about radiocarbon and thermal analysis, isotopic partitioning procedures and quantification of their uncertainty, and statistical analyses can be found in Supplementary Methods.

### Study soils, experimental design and soil sampling
We selected three soil types: eutric cambisol, chromic vertisol and silandic andosol[70]. The three soil profiles studied were found in long-term semi-natural grasslands located relatively close to each other (<100 km) in Auvergne, France. They developed under a similar temperate semi-continental climate and mainly differed by their parent materials: granite, basalt, and trachyandesite for the cambisol, vertisol and andosol, respectively (Supplementary Table 7). The three soil profiles studied were well-aerated since none showed reductimorphic layers, and redoximorphic features were present only in the vertisol.

The experiment had a factorial design of two crossed factors: two soil layers including topsoil *versus* subsoil, and three soil types for a total of six treatments each including four replicates. We collected 20 cm high soil cores from the two layers for each soil profile. Intact soil columns of 8 cm diameter were extracted using a percussion core drill that can be opened from sideways. Topsoil cores were taken in the 5–25 cm depth part of the A horizon, which allowed us to remove both the native vegetation and a large proportion of their fresh litter. Subsoil cores were taken in the top 20 cm of the B horizon, that are at 40–60, 55–75 and 35–55 cm depth respectively for the cambisol, vertisol and andosol (Supplementary Fig. 5).

### Soil biogeochemical analyses
Four soil cores of each treatment were used to characterise soil biogeochemical properties. These soil cores were first sieved at 2 mm and a portion was air-dried at room temperature. A portion of the air-dried soil was also ground (<250 µm) to homogeneity.

Soil C and N concentrations and $\delta^{13}C$ was determined on ground soil containing ~1 mg C using an elemental analyser (EA, Carlo Erba, Rodana, Italy) coupled to an isotope-ratio mass spectrometer (IRMS; Elementar, Langenselbold, Hesse, Germany). None of the soil profiles contained carbonates, and total soil C content is interpreted as SOC only here. To determine the relative contribution of mineral-associated organic matter *versus* particulate organic matter to SOC, each soil sample was fractionated by particle size (50 µm) after full soil dispersion[71]. Briefly, 5 g of air-dried soil was shaken for 18 h in sodium hexametaphosphate (0.5%) with beads to completely disperse the soil. The dispersed soil was then rinsed onto a 50 µm sieve and the fraction passing through (<50 µm) was collected as MAOM, while the fraction remaining on the sieve was collected as POM. Each fraction was then analysed for C concentration using an EA.

The radiocarbon signature ($\Delta^{14}C$) of SOC was determined by Accelerator Mass Spectroscopy (AMS) on ground soil containing respectively ~1 and 0.14 mg C for topsoil and subsoil with a Mini Carbon Dating System (*ECHo*MICADAS) operated at LSCE (Climate and Environment Sciences Laboratory) in Gif-sur-Yvette, France. To assess SOC dynamics, we estimated SOC turnover time based on radiocarbon measurements using a modelling approach[72,73]. The following time-dependent, homogeneous one-pool model was used:

$$F^{14_C}_{SOC,t} = \frac{1}{\tau}F^{14_C}_{atm,t} + F^{14_C}_{SOC,t-1}\left(1 - \frac{1}{\tau} - \lambda\right) \quad (1)$$

given $F^{14_C} = \left(\dfrac{\Delta^{14}C}{1000}\right) + 1$

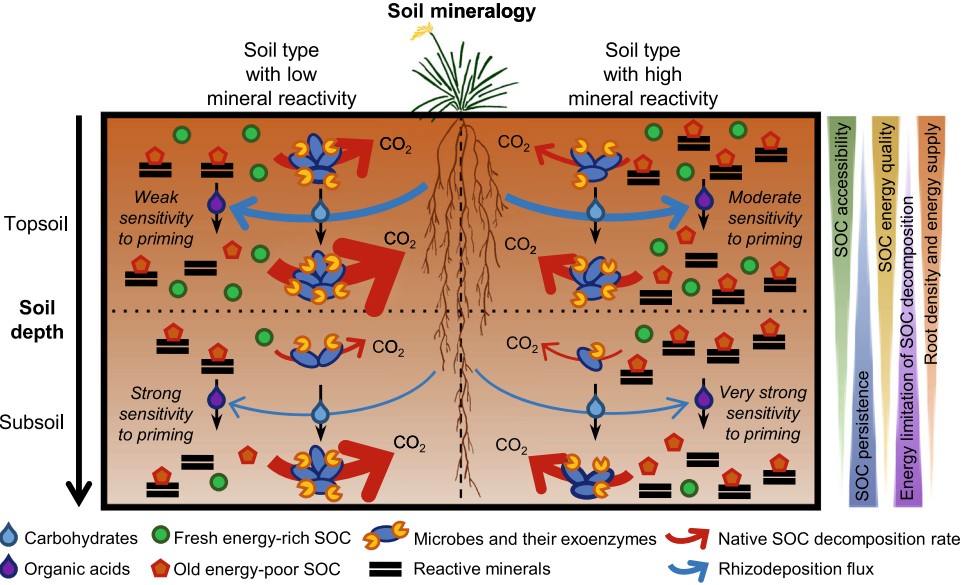

**Fig. 6 | Conceptual model illustrating the bioenergetic control of SOC dynamics and its sensitivity to rhizosphere priming across soil depth and mineral reactivity contexts.** The width of native SOC decomposition arrows is proportional to its rate. The rhizosphere priming intensity represents the magnitude in the acceleration of native SOC decomposition induced by a given density and energy supply of plant roots. Plant image by Alice Trotel.

where $t$ is the time (in year), $F^{14C}_{SOC}$ is the $^{14}C$ content of SOC, $F^{14C}_{atm}$ is the $^{14}C$ content of $CO_2$ in the local atmosphere, $\tau$ is the mean SOC turnover time and $\lambda$ is the radioactive decomposition constant for $^{14}C$ ($1.21 \times 10^{-4}$ year$^{-1}$). We ran the model from 50 kyr BP until the year 2016 to calculate the predicted SOC $\Delta^{14}C$ at the year of sampling for a range of $\tau$ values (1 to 30,000 years). We then derived $\tau$ values from our $\Delta^{14}C$ measurements for each sample based on the relationship between $\tau$ and predicted $\Delta^{14}C$ (Supplementary Fig. 6). Though the model assumption of SOC as a homogeneous one-pool is clearly an oversimplification, our approach remains useful to compare SOC dynamics across soil samples[9].

Thermal analyses were used to characterise the bioenergetic signature of SOM. The activation energy ($E_a$) of SOC thermal decomposition was measured by evolved gas analysis during ramped combustion using Rock-Eval® thermal analysis. Sequential ramping by pyrolysis and oxidation was performed on ~60 mg of ground soil using a Rock-Eval® 6 Turbo device (Vinci Technologies, France). Hydrocarbon effluents were quantified by flame ionisation detection during the pyrolysis phase, while CO and $CO_2$ were quantified by infra-red detection during both ramping phases. Based on evolved SOC kinetics, a regularised inverse method was used to determine the continuous distribution of $E_a$ that best predicts the profile of SOC decay measured during ramped combustion[74] (Supplementary Fig. 1a, b). The distribution of $E_a$ was then integrated to calculate the mean ($\mu E_a$) and standard deviation ($\sigma E_a$) of activation energy. Additionally, the temperature at which 90% of hydrocarbons was evolved during pyrolysis ($T_{90}$-$H_xC_y$-pyrolysis, °C), and the temperatures at which 50% of $CO_2$ was evolved during pyrolysis and oxidation ($T_{50}$-$CO_2$-pyrolysis and $T_{50}$-$CO_2$-oxidation, °C) were used as indices of SOC thermal stability[75]. The hydrogen index (HI) and oxygen index (OI) providing information about elemental H:C and O:C ratios of SOM were calculated respectively as the amount of hydrocarbons and the amount of $CO_2$ and CO formed during pyrolysis divided by SOC concentration.

Energy density of SOC ($\Delta E$) was measured by differential scanning calorimetry (DSC)[28,76]. Oxidation ramping was performed on ~60 mg of ground soil using a DSC thermal analyser (TGA-DSC 3+ model, Mettler-Toledo, Greifensee, Switzerland) to measure the net energy released by SOM combustion (enthalpy of combustion), knowing that some of the energy applied to the sample is consumed by the breakdown of the

organo-mineral associations. The $\Delta E$ was calculated as the net energy released determined by integration of heat flux over the exothermic region associated to SOC combustion (185–600 °C, Supplementary Fig. 1c), divided by SOC concentration and multiplied by SOC molar mass (Supplementary Table 8, Supplementary Methods). Calorimetry also allows to estimate the degree of reduction of SOC ($\gamma_{SOC}$)[77], which was calculated here as $\Delta E$ divided by $Q_0$, the oxycaloric quotient representing the ratio between the enthalpy of combustion and the degree of reduction[76]. We used a $Q_0$ value of 109.04 kJ mol$^{-1}$ SOC degree of reduction$^{-1}$, obtained from the average of the heat of combustion of a large set of organic compounds[78]. Additionally, a return-on-energy-investment (ROI) parameter was calculated as $\Delta E$ divided by $\mu E_a$[26] and was used here as an index of SOC energy quality[27,28].

Soil microbial biomass was measured on fresh sieved soil by the chloroform-fumigation-extraction method[79]. We performed an extraction with 50 mL 0.5 M $K_2SO_4$ on 10 g of fresh soil. A second set of samples was placed in a vacuum desiccator and fumigated with chloroform for 24 h prior to $K_2SO_4$ extraction as above. Extractable C was analysed using an automated analyser (TOC-L analyser, Shimadzu, Milton Keynes, UK). Microbial biomass C was calculated as the difference between the fumigated and unfumigated C extracts and a correction for extraction efficiency was applied by dividing with a coefficient of 0.45[80]. Initial soil mineral N content was measured from 25 g of fresh soil after extraction in 2 M KCl using a continuous-flow analyser (AA3, Bran + Luebbe, Norderstedt, Germany). All roots retained by sieving of soil cores and visible roots in sieved fresh soil were handpicked and washed with tap water. After drying at 60 °C for 48 h, roots were weighted and root density was calculated as root dry mass divided by the soil core volume.

We also characterized the mineral composition of soil samples. Soil pH$_{water}$ was measured in a 1:5 soil:solution ratio after 1-h end-over-end shaking. Soil clay content was measured using the pipette method[81]. Phyllosilicate mineralogy was determined by X-ray diffraction[82]. Random oriented powders with a Philips PW 3710 X-ray diffractometer with Cu-Kα radiation at 40 kV and 40 mA were used to obtained spectra (Supplementary Fig. 7). A counting time of 13 s per 0.02° step was used for 2θ in the range 3.5–80°. Cation exchange capacity (CEC) as well as major exchangeable cations (Ca, Mg, Na, K, Al, Mn, Fe) were determined determined using the cobalt hexamine

exchange method[83]. The concentrations of divalent cations involved in organo-mineral cation bridging was calculated as the sum of exchangeable calcium and magnesium ($Ca_{ex}+Mg_{ex}$). Pedogenic reactive metals were quantify by selective dissolution procedures of Fe, Al and Si using standard methods of citrate–dithionite (d), acid ammonium oxalate (o), and sodium pyrophosphate (p) extractions[84]. Crystalline Fe minerals was then calculated as the difference between $Fe_d$ and $Fe_o$ ($Fe_{d-o}$). We used $Fe_o$ to quantify short-range-ordered Fe-oxyhydroxides such as ferrihydrite, while $Al_o+Si_o$ were used to quantify organo-metal complexes and short-range-ordered aluminosilicates such as allophane. $Al_p$ allows to quantify organo-metal complexes, but pyrophosphate is not completely selective and can also extract Al from silicates, which could be indicated by $Si_p$[84]. Organo-metal complexes was thus calculated as $Al_p$-$xSi_p$, where $x$ is the Al:Si ratio of the dominant clay mineral ($x = 0.5$ for subsoil of the vertisol dominated by montmorillonite, and $x = 1$ for all other layers dominated by vermiculite, kaolinite or halloysite).

### Plant isotopic labelling and soil incubation experiment

Soil cores collected for the experiment were immediately proceeded in the field following sampling to establish microcosms with a new soil column made of intact soil cores derived exclusively from either topsoil or subsoil (Supplementary Fig. 2). For each microcosm, three soil cores of the same layer were gently stacked vertically and tightly sealed together within a polyethylene sheath before transfer into a 60 cm high PVC pot (diameter 10 cm, height 60 cm, with a permeable bottom-cap to allow drainage). For each the six treatments including the three soil types and the two layers, we included four planted replicates and four unplanted controls for a total of 48 microcosms. The experiment started within 2 months after sampling and microcosm were stored at 4 °C until then.

The experiment was performed for 279 days, from late August 2016 until early July 2017. Two weeks before starting the incubation experiment, the microcosms were transferred at ambient temperature, irrigated until water saturation and weighed after 48 h of water percolation to measure the soil water-holding capacity (WHC). Planted microcosms were sown (1400 seeds $m^{-2}$) with *Dactylis glomerata*, a fast-growing grass species with a dense and deep root system commonly found in temperate grassland[85]. After germination, microcosms were transferred to a greenhouse exposed to natural light and temperature conditions (Clermont-Ferrand, temperate semi-continental climate). The greenhouse was coupled to a continuous dual-labelling ($^{13}C/^{14}C$) system[86,87]. Labelled air depleted in both $^{13}C$ and $^{14}C$ was produced by injecting fossil fuel-derived $CO_2$ ($\delta^{13}C = -35.23 \pm 0.02$ ‰, $\Delta^{14}C$-0‰) in $CO_2$-free air ([$CO_2$] < 20 ppm) up to reach ambient $CO_2$ concentration (400 ppm), and the greenhouse was continuously supplied with labelled air during daytime, with a flow renewing the greenhouse volume once every 2 min to maintain constant $CO_2$ concentration, $\delta^{13}C$ and $\Delta^{14}C$[86]. Soil water content was monitored daily using soil moisture sensors (ECH2O EC-5, Decagon®, USA) inserted at 5 cm and drip irrigation was adjusted individually for each treatment as to maintain moisture around $85 \pm 5\%$ of WHC. In order to compensate for the low nutrient availability and plant growth potential expected in subsoil relative to topsoil, we fertilised the planted subsoil microcosms for each soil type on day 51 after planting. Unplanted subsoil microcosms were kept unfertilised to avoid excessive concentrations of mineral nutrients, which already tend to accumulate in soils in absence of rhizodeposition and nutrient uptake by plant roots[41,88]. The fertilisation solution was composed of inorganic N ($NH_4NO_3$), P, S, K and Mg, with a dose of 11.5, 0.6, 1.0, 1.5 and 0.7 g $m^{-2}$, respectively. The fertilization was adjusted to compensate for initial differences in measured soil mineral N concentrations compared with planted topsoil microcosms, with a single dose added for each soil type and a second dose including only N applied to the andosol (Supplementary Table 9). This allowed to reach similar soil mineral N concentrations, plant N

concentration and biomass between planted topsoil and subsoil microcosms following fertilization (Supplementary Tables 9, 10).

For the first incubation series, we measured $CO_2$ fluxes of each microcosm 76, 139, 174, 201, 242 and 272 days after planting, from late fall until early summer ($n = 288$ incubations). Microcosms were sealed in opaque airtight chambers and incubated for 24 h at 21.5 °C. Chamber gas was sampled at the end of incubation, and its $CO_2$ concentration and $\delta^{13}C$ were measured using a gas chromatograph (Clarus 480, Perkin Elmer, Waltham, MA, USA) and an isotopic analyser (G2201-i, Picarro, Santa Clara, CA, USA). The amount and $\delta^{13}C$ of $CO_2$ derived from plant-soil respiration were corrected for background atmospheric $CO_2$.

For the second incubation series performed 279 days after planting, the soil column was gently extracted from each microcosm. Shoots were cut at the soil surface for planted microcosms, and the soil column was vertically sliced to recover the three original soil cores. Each soil core was transferred into a 3 L flask as gently as possible to maintain the soil core structure intact ($n = 144$ incubations). After a preincubation period of 24 h at 21.5 °C, flasks were airtight-sealed and incubated for 24 h at 21.5 °C. The evolved $CO_2$ was trapped in NaOH and its concentration was measured using an automated analyser (TOC-L analyser, Shimadzu, Milton Keynes, UK). After carbonate precipitation with $BaCl_2$ and filtration, the $\delta^{13}C$ of evolved $CO_2$ was analysed by an EA-IRMS. We also measured the $\Delta^{14}C$ of evolved $CO_2$ by AMS analyses as described above for planted subsoil cores of 0–20 and 40–60 cm depth, and unplanted subsoil cores of 40–60 cm depth. Given the high cost of AMS analyses, $\Delta^{14}C$-$CO_2$ measurements were restricted to the cambisol and andosol that a priori featured the most contrasting mineral reactivity ($n = 24$ incubations).

After the incubation, each planted soil core was sieved at 2 mm. Roots retained by sieve and all visible roots in sieved soil were hand-picked and washed with tap water. Shoot, root and soil materials were dried at 60 °C, weighed, grounded and analysed separately for C and N concentrations and $\delta^{13}C$ using an EA-IRMS as described above. For the cambisol and andosol, $\Delta^{14}C$ of root biomass in subsoil cores was measured as described above.

### Isotopic partitioning and calculations

For both incubation series, the continuous isotopic labelling of plants with $^{13}C$-depleted air allowed us to partition total respiration into its soil and plant/root sources (~25‰ average difference in $\delta^{13}C$). It was calculated using the following equations:

$$R_{soil} = R_{total} \times \frac{\delta^{13}C_{total} - \delta^{13}C_{plant}}{\delta^{13}C_{soil} - \delta^{13}C_{plant}} \tag{2}$$

$$R_{plant} = R_{total} \times \frac{\delta^{13}C_{total} - \delta^{13}C_{soil}}{\delta^{13}C_{plant} - \delta^{13}C_{soil}} \tag{3}$$

where $R_{total}$ and $\delta^{13}C_{total}$ are respectively the total $CO_2$ flux and its $\delta^{13}C$ from plant/root-soil respiration at the microcosm/core scale; $R_{soil}$ and $\delta^{13}C_{soil}$ are respectively the $CO_2$ flux and its $\delta^{13}C$ from microbial respiration of unlabelled native (pre-existing) SOC and root litter; and $R_{plant}$ and $\delta^{13}C_{plant}$ are respectively the $CO_2$ flux and its $\delta^{13}C$ from respiration of labelled (recently fixed) plant/root-derived organic carbon (OC). For the first incubation series, we used as $\delta^{13}C_{plant}$ the mass-weighted $\delta^{13}C$ of the mesocosm shoot and living root biomass, assuming negligible $^{13}C$ fractionation during whole-plant respiration[89]. A parallel experiment running during the same period and using a common labelling system found that the $\delta^{13}C$ of plant biomass was constant through time[86], ensuring that the labelling remained homogeneous throughout the experiment. For the second incubation series, we used as $\delta^{13}C_{plant}$ the $\delta^{13}C$ of living root biomass corrected by a $\delta^{13}C$ fractionation factor of root respiration, which was assumed to be

−0.61‰ for grass species based a previous study[87]. For both series of incubations, we used as $\delta^{13}C_{soil}$ the $\delta^{13}C$ of $CO_2$ derived from the respiration of native SOC in unplanted controls.

In the second incubation series, the continuous dual-labelling ($^{13}C$/$^{14}C$) of plants allowed us to quantify the radiocarbon signature of native SOC respired by decomposers ($\Delta^{14}C_{soil}$) by partitioning the $\Delta^{14}C$ signature of total respiration into its soil and root sources[17,20]. It was calculated using the following equation:

$$\Delta^{14}C_{soil} = \frac{R_{total} \times \Delta^{14}C_{total} - R_{plant} \times \Delta^{14}C_{plant}}{R_{soil}} \quad (4)$$

where $R_{total}$ and $\Delta^{14}C_{total}$ are respectively the total $CO_2$ flux and its $\Delta^{14}C$ of root-soil respiration at the core scale, $\Delta^{14}C_{plant}$ is the $\Delta^{14}C$ of root-derived OC respiration, and $R_{plant}$ and $R_{soil}$ are the $CO_2$ fluxes of respectively root-derived and soil-derived OC respiration as calculated in Eqs. 2, 3. Assuming no fractionation of $^{14}C$ during root respiration, we used the $\Delta^{14}C$ of living root biomass as $\Delta^{14}C_{plant}$.

As the root material harvested for topsoil was composed of both pre-existing root litter (unlabelled) and living roots (labelled) that could not be clearly visually sorted, we used an isotopic partitioning method to estimate living root biomass ($Root_{living}$) for each planted topsoil core using the following equation:

$$Root_{living} = Root_{total} \times \frac{\delta^{13}C_{total} - \delta^{13}C_{dead}}{\delta^{13}C_{living} - \delta^{13}C_{dead}} \quad (5)$$

where $Root_{total}$ and $\delta^{13}C_{total}$ are respectively the biomass and $\delta^{13}C$ of both dead and living roots; and $\delta^{13}C_{living}$ and $\delta^{13}C_{dead}$ are the $\delta^{13}C$ of respectively living and dead roots.

We also quantified the net rhizodeposition corresponding to root-derived OC remaining in the soil after microbial utilisation. It was estimated for each planted soil core using the following equation:

$$Net\ rhizodeposition = SOC_{total} \times \frac{\delta^{13}C_{SOC-final} - \delta^{13}C_{SOC-initial}}{\delta^{13}C_{root} - \delta^{13}C_{SOC-initial}} \quad (6)$$

where $SOC_{total}$ and $\delta^{13}C_{SOC-final}$ are respectively the concentration and $\delta^{13}C$ of SOC in planted soil core at the end of the experiment, $\delta^{13}C_{SOC-initial}$ is the average $\delta^{13}C$ of SOC from initial soil cores, and $\delta^{13}C_{root}$ is the $\delta^{13}C$ of living root biomass in planted microcosm.

In order to evaluate the uncertainty associated with our isotopic mixing model assumptions, we performed sensitivity analyses where we quantified the error in source proportion related to a 1‰ variation in the $\delta^{13}C$ of both sources for each isotopic partitioning (Supplementary Methods). Low levels of uncertainty were found for every isotopic partitionings, providing evidence that our results were robust (Supplementary Tables 10–14, see Supplementary Methods for further details).

Assuming first-order decomposition kinetics, native SOC decomposition rate ($k_{SOC}$) was calculated as native SOC respiration ($R_{soil}$) divided by initial SOC concentration. The corresponding rhizosphere priming effect (RPE, in % of change in $k_{SOC}$ for planted relative to unplanted soils) was calculated using the following equation:

$$RPE = \frac{k_{SOC\ planted} - k_{SOC\ unplanted}}{k_{SOC\ unplanted}} \times 100 \quad (7)$$

The respiration of root-derived OC ($R_{plant}$) in the second incubation series includes both the autotrophic respiration of roots and the heterotrophic respiration of soil microbial decomposers derived from rhizodeposits. We thus used it here as a proxy of plant C allocation belowground to root activity, including root maintenance, growth and uptake of water and nutrients, as well as rhizodeposition of fresh OC to soil which is then quickly taken up and metabolised by rhizosphere

microbes[33]. This allowed us to assess how root effects on SOC decomposition could depend on plant inputs to soil of energy that could take different forms, including food substrates for microbes (metabolic energy), ligand exudates desorbing SOC from minerals (chemical energy) and root uptake of water and nutrients breaking up soil aggregates by physical disturbances (physical energy). After converting $R_{plant}$ from an amount of C in $CO_2$ (mg) into an amount of glucose (mol), we calculated the plant energy investment into root activity (J dm$^{-3}$ day$^{-1}$) as $R_{plant}$ multiplied by the heat of combustion of glucose, that is 2807 kJ per mol[78]. This estimation relies on the assumption that glucose composed most of the fresh photosynthate-C metabolised by both roots and soil microbes[32].

### Statistical analyses

All analyses were performed using R v3.4.3. We used a rotated principal component analysis (rPCA) to explore soil properties covariance and divergence between treatments. Analyses of variance were used to partition the variance explained by factors 'soil layer', 'soil type' and their interaction for the two first axis scores and soil properties.

For each incubation series, the responses of $k_{SOC}$, RPE and $\Delta^{14}C_{soil}$ to predictors were assessed by regression for each treatment. We tested linear ($Y = a + bX$), polynomial ($Y = a + bX + cX^2$) and power ($Y = aX^b$) regression functions, where Y is the response variable and X is the predictor. The $k_{SOC}$ and RPE values were standardised for each treatment to a common high value of the following predictors: 'respiration of plant-derived OC' for the first incubation series, 'living root density' and 'plant-derived OC respiration' for the second incubation series. Additionally, we used analyses of covariance including the quantitative explanatory variables, the factors 'soil layers' and 'soil type', and their interactions as fixed factors to test their effects and quantify the proportion of variance they explain. To deal with the repeated measures design in both incubation series, we used linear mixed-effect models including 'microcosm' as random factor in regression and analyses of covariance.

We explored bivariate relationships between variation in SOC dynamics and SOM properties across depth using partial regression and correlation analyses controlling for soil types. Additionally, we performed an ordination of SOC dynamics variables constrained by soil biogeochemical predictors using a redundancy analysis.

### Reporting summary

Further information on research design is available in the Nature Portfolio Reporting Summary linked to this article.

## Data availability

All data generated or analysed during this study are provided in the Supplementary Information (Supplementary Data). Additional data that support the findings of this study are available from the corresponding author (L.H.) upon request. Source data are provided with this paper.

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

## Acknowledgements

This work is dedicated to Jerôme Balesdent, Research Director at INRAE, who died prematurely at the age of 63. This study was financially supported by the ANR project DEDYCAS (ANR 14-CE01-0004) led by J. Balesdent, and the Programme EJP Soil of the European Commission (project AgroSeqC). We thank S. Revaillot, F. Savignac, L. Andanson, and A. Salcedo for their technical support. We also thank V. Genevois, K. Klumpp, F. Louault, and J. Reymond for their help in soil sampling and

access to the sites, including the AnaEE-France (ANR-11- INBS-0001) in natura SOERE ACBB permanent grassland experimental sites of Theix and Laqueuille. We thank the certified facility in Functional Ecology (PTEF OC 081) from UMR 1137 EEF and UR 1138 BEF in research centre INRAE Nancy-Lorraine for its contribution to isotopic analysis of plant and soil samples. The PTEF facility is supported by the French National Research Agency through the Laboratory of Excellence ARBRE (ANR-11-LABX-0002-01). The TGA-DSC thermal analyser (ISTerre) was partially funded by a grant from Labex OSUG@2020 (Investissements d'Avenir, ANR10-LABX56).

## Author contributions

L.H. and S.F. conceived the ideas and designed methodology. L.H., G.A. and S.F. performed the soil sampling. L.H. and S.F. performed most of the soil biogeochemical analyses, except for radiocarbon analyses performed by C.H., Rock-Eval thermal analyses performed by P.B., F.B. and L.C., DSC thermal analyses performed by A.F.M., and X-ray diffraction and pedogenic reactive metals analyses performed by I.B.D. and J.B. L.H. and S.F. performed the sampling and all analyses of $CO_2$ during the incubations, except for radiocarbon analyses performed by C.H. L.H. analysed the data and led the writing of the manuscript. All authors contributed critically to the drafts and gave final approval for publication.

## Competing interests

The authors declare no competing interests.
