## [Peer Review File · Nature Communications]

Bioenergetic control of soil carbon dynamics across depthReviewer #1 (Remarks to the Author):

This is a really fascinating topic and I agree the working out the stability of SOC at depth is crucial. I very much enjoyed reading the intro, although I felt in places the thoughts were not fully formed (see below).

Generally, the experimental setup needs to be much clearer. I understand why a lot must go into the supplementary, but it isn't clear to me which sections of the methods correspond to the field and the incubations (the field is not really mentioned in the methods or supplementary, just the intro).

On a similar note, in the methods there are sections in the main text that do not tell the reader how much soil was used, dry or wet, and what machine was used to determine the values. In particular the Soil geochemical properties section (line 549) has no reference to the method used and is lacking all the information I have mentioned. Please could you go through and make sure for the methods that has made it in, there is sufficient information to understand broadly what was done?

I'm really sorry about this next point, but I'm having problems differentiating between the different reds in the figures (cambisol and vertisol especially). The green is easier although still not very distinguishable. My suggestion would be to have each soil type the same colour and have filled/empty symbols for the topsoil/subsoil. And dashed or solid lines. I'm not colourblind, but this is a problem for me and likely to be worse for certain others.

Additionally, figure 1 is very busy. It is interesting, but there is a lot going on. Could you move the mean SOC age inset somewhere else? This is also true of figure 3. There is so much going on it confuses the eye, and the red green colourscheme is difficult because you automatically compare the reds and the greens, where actually for panel a I realised I should probably be looking at the dashed vs the solid lined. I think pull out the insets and put them beneath the decomp graphs as a set, make the legend bigger and more obvious, and change the colourscheme or presentation of planted/unplanted because it conflicts with the topsoil/subsoil.

I am surprised you did not include any microbial diversity data to pair with the isotopic data. Is this in the pipeline in a different paper? I think it would strengthen your points somewhat to be able to directly show microbial community or enzymatic activity rates.

Line 61: half of SOC- do you mean locally? Surely this cannot be a general overall estimate, as many soils are shallower than 30cm? Could you clarify?

Lines 69-73: you don't mention abiotic limitations on biological activity at depth- low temperatures, increasingly anaerobic conditions. Do you not consider these to be important factors in protection of SOC in deep layers?

Line 90: why might this be? Do molecules that escape rapid metabolism get washed downwards through the soil profile? Is there evidence for this?

Line 110: how deep are we talking? Again, are the abiotic variables like temperature and aeration not a consideration?

Lines 442: What happened to the 25-40cm deep soil? Does this not represent an interesting transition zone?

Line 445: this is rather hot, nitrogen based molecules are likely to be inaccurate much above 70 °C. Using 105 °C comes with the caveat that organic matter is likely to be lost (Reynolds 1970, Journal of Hydrology, 11, 258-273; Lekshmi et al. 2014, Measurement, 54, 92-105).

Line 280: this is the first time this point has come up, and I think it needs to be dealt with in the intro or people will be wondering why (like I did).

Supplementary lines 200-206: I am concerned about the addition of fertiliser to only the planted soils- would this not lead to accelerated priming effects? If so, the planted and unplanted treatments are not comparable.

Reviewer #2 (Remarks to the Author):

Manuscript NCOMMS-22-26671 tests a novel bioenergetics-based framework for soil organic matter persistence as a function of soil depth. The dynamics and persistence of soil organic matter has received a great deal of attention over the past 20+ years. The community has undergone a paradigm shift in understanding the major controls on soil organic matter persistence - from chemical composition of recalcitrant moieties, to mineral association and ecosystem properties. These are well synthesized by Schmidt et al 2011, cited in the manuscript. The challenge from the new paradigm, however, has been how to make the proposed, emergent property of ecosystem-level persistence measurable, modelable, and predictable. And hence it appears that a new paradigm is being developed, based on bioenergetics and rooted in thermal analysis techniques. This manuscript represents an important contribution to this new paradigm development, with strong conceptual underpinnings and appropriate and rigorous experimental design and data.

The experimental study is well designed. All methods are appropriate, and calculations appear to be correct and valid. The combination of multiple thermal analyses methods, along with incubations and isotopic analyses make for a robust set of observations that support the conclusions drawn by the authors. When not necessarily creating an extensively comprehensive dataset (as would a meta-analysis of a large, global dataset), the inclusion of soils with differing mineralogy (ie, surface reactivity) makes the observations even more robust. Reporting and visualization of the results is highly appropriate, with some caveats:

- 1. In Table 1, "Depth effect" might be better reported as an actual statistical outcome. That is, rather than a mean relative/absolute change, it could be reported as the ANOVA effect size (which can be calculated multiple ways). Alternatively, a statistical test (p-value, etc.) might be reported to demonstrate the degree to which the difference falls outside the range attributable to natural variability.**
- 2. There are a large number of loading arrows in Fig 2a. Perhaps only show those that are "significant" in that they are deemed the most important and worth of explanation in the main body of the text.**
- 3. The caption refers to the width of the decomposition arrow being proportional, but the differences in width are perhaps too subtle to be immediately noticeable. Perhaps elements of the cartoon can be further exaggerated to highlight the points being made.**

The manuscript is very well written and I have very few, very minor editorial comments (see below). However, I did find that a large portion of the discussion section (lines 245 through 343) is largely an elaboration of what is in the results section. Perhaps it's because the results section is written sufficiently clearly and understandably, the first part of the discussion seems somewhat redundant. Going forward from line 344 is what I consider to be a discussion in the conventional sense. Perhaps the discussion section as a whole can be shortened for conciseness. Alternatively, the goals of the elaboration of the results can be to place observations in context with other values reported in the literature rather than a repetition of what is in the results section. I understand there are likely few reported data to compare to, as this is an evolving paradigm with novel experiments. Another possibility would be to elaborate on the conceptual framework (both historically as it seems related to the early ecosystem work of Odum) and then place the observations within this framework, rather than to elaborate on the results themselves.

In terms of editorial comments, the text is largely free of grammatical and linguistic errors. However, I would recommend that the terms "higher" and "lower" be reserved for vertical position since the study involves samples taken higher and lower in the soil profile. Instead, these terms should be replaced in comparisons with terms that are more

directly equivalent to ">" and/or "<", such as "larger/smaller", "greater/lesser", etc. Also...

In46: "reactivity contexts" in stead of "context"

In60: "30 cm deep" instead of "depth"

In101: "To date" instead of "So far"

In106 and throughout: The term "field" implies measurement potential performed "in-situ" and literally in the field. I think this term can be omitted in most usages in the manuscript.

In109: I don't think "genericity" is a word, and even if so, it's awkward. Replace with "robustness"

In128: "Microbial biomass" instead of "biomasses"

In162: "in soil columns" instead of "on soil columns"

In167: "consisted of" instead of "in"

In218: "high amounts" instead of "amount"

In219: "mineral composition" instead of "nature"

In219: omit "indeed"

In226: omit "in contrast"

In351: "have often" instead of "has often"

Overall, I find the manuscript to be of sufficient rigor, quality and novelty for publication in Nature Communications with only minor revisions.

Prof. Alain Plante
University of Pennsylvania

REPLY TO REVIEWERS

Manuscript NCOMMS-22-26671

Reviewer #1 (Remarks to the Author):

This is a really fascinating topic and I agree the working out the stability of SOC at depth is crucial. I very much enjoyed reading the intro, although I felt in places the thoughts were not fully formed (see below).

Comment: Generally, the experimental setup needs to be much clearer. I understand why a lot must go into the supplementary, but it isn't clear to me which sections of the methods correspond to the field and the incubations (the field is not really mentioned in the methods or supplementary, just the intro).

Response: *To improve clarity, we reformatted the structure of the methods section so to have one subheading specifically dedicated for each of these two parts of our study (See L. 394 to 547).*

Comment: On a similar note, in the methods there are sections in the main text that do not tell the reader how much soil was used, dry or wet, and what machine was used to determine the values. In particular the Soil geochemical properties section (line 549) has no reference to the method used and is lacking all the information I have mentioned. Please could you go through and make sure for the methods that has made it in, there is sufficient information to understand broadly what was done?

Response: *The methods section has been complemented with the additional information you suggested, especially for soil geochemical properties which are fully located in the main text manuscript instead of the supplementary. More methodological details about the plant isotopic labelling and soil incubation experiment are also now provided in the main text instead of the supplementary.*

Comment: I'm really sorry about this next point, but I'm having problems differentiating between the different reds in the figures (cambisol and vertisol especially). The green is easier although still not very distinguishable. My suggestion would be to have each soil type the same colour and have filled/empty symbols for the topsoil/subsoil. And dashed or solid lines. I'm not colourblind, but this is a problem for me and likely to be worse for certain others.

Response: *We modified the figures according to your suggestion to improve their readability.*

Comment: Additionally, figure 1 is very busy. It is interesting, but there is a lot going on. Could you move the mean SOC age inset somewhere else?

Response: *The information in the inset about mean SOC age has been moved in Table 1. To improve the figure readability, we reduced the number of variables in the ordination from 25 to 18.*

This is also true of figure 3. There is so much going on it confuses the eye, and the red green colourscheme is difficult because you automatically compare the reds and the greens, where actually for panel a I realised I should probably be looking at the dashed vs the solid lined. I think pull out the insets and put them beneath the decomp graphs as a set, make the legend bigger and more obvious, and change the colourscheme or presentation of planted/unplanted because it conflicts with the topsoil/subsoil.

Response: *All the modifications you suggested have been implemented in the current version of the figs.3, 4 and 5.*

Comment: I am surprised you did not include any microbial diversity data to pair with the isotopic data. Is this in the pipeline in a different paper? I think it would strengthen your points somewhat to be able to directly show microbial community or enzymatic activity rates.

Response: *We have not measured microbial diversity or enzymatic activities in this study for several reasons. The depth-dependency of the diversity and structure of microbial communities is clearly an interesting question, but has already been the subject of excellent publications¹⁻⁴. We also believe this is a peripheral question that is not essential to test our main hypothesis, and the significant investment*

we made into the energetic aspect central to our study did not allowed us to address this question. Enzymatic activity data would have been very relevant to our question but measuring in situ enzymatic activity rates in soil is very challenging. Standard methods usually rely on addition substrates and thus rather reflect potential enzymatic activities which may not reflect in situ patterns since substrate availability in real-world soils is notoriously limited at depth.

Comment: Line 61: half of SOC- do you mean locally? Surely this cannot be a general overall estimate, as many soils are shallower than 30cm? Could you clarify?

Response: *No, we are actually talking about global SOC stocks. This has been clarified here:*

*L. 59-60. “a major portion corresponding to around half of **the global SOC stock** is stored in deeper soil layers (subsoil)^{5,6}.”*

This is based on the most up to date global soil organic carbon (SOC) inventory⁶, which by the way showed in its Fig. 15 that the vast majority of world soils are more than 50 cm deep.

Comment: Lines 69-73: you don't mention abiotic limitations on biological activity at depth- low temperatures, increasingly anaerobic conditions. Do you not consider these to be important factors in protection of SOC in deep layers?

Response: *We added a new sentence in the introduction to explicitly consider the importance of these factors in deep SOC persistence:*

*L. 74-77. “**Suboptimal environmental conditions such as low temperature and anaerobic conditions have additionally been mentioned^{7,8}, and obviously represent key drivers of deep SOC persistence for permafrost and peatlands. Empirical evidence supporting their importance for mineral well-aerated soils remains however limited⁹⁻¹¹.**”*

Since our study does not include specific soil types such as permafrost and peatlands/wetlands for which climatic factors are important drivers of deep SOC persistence, we also modified this sentence at the end of the introduction so to circumscribe the scope of inference of our study:

*L. 107-109. “We thus investigated here the bioenergetic control of the depth-dependency of SOC dynamics **in temperate well-aerated mineral soils.**”*

We also added this sentence in the methods section to be clarified that three soil profiles are all well-aerated:

*L. 383-384. “**The three soil profiles studied were well-aerated since none showed reductimorphic features, and redoximorphic features were present only in the vertisol.**”*

Comment: Line 90: why might this be? Do molecules that escape rapid metabolism get washed downwards through the soil profile? Is there evidence for this?

Response: *There studies providing evidence that molecules percolate downwards through the soil profile and contribute to the formation of deep SOC in addition to inputs from plant roots^{12,13}. This comment points that it could be important to briefly mention the factors contributing to deep SOC formation, so we added a new sentence about this in the beginning of the introduction:*

*L. 60-63. “**Deep SOC formation can derive from dissolved organic matter and colloidal organo-mineral particles percolating downward through the soil profile, as well as organic matter inputs by deep plant roots, accumulation of eroded soil downhill and bioturbation by soil fauna such as earthworms^{5,12-14}.**”*

Comment: Line 110: how deep are we talking? Again, are the abiotic variables like temperature and aeration not a consideration?

Response: *Deep SOC refers to SOC located at depth greater than 30 cm, which is how deep SOC is already defined in the beginning of our introduction. Abiotic variables like temperature and oxygen availability are indeed not our main consideration, and our study was not specifically designed to test*

the importance of these factors. Since our results provide however some interesting clues about the importance of these factors, we now dedicate a specific paragraph to this question in the discussion:

L. 311-322 *“Suboptimal environmental conditions have been mentioned as potentially important driver of deep SOC persistence^{7,8}. Our results largely confirmed empirical studies showing that their importance for mineral well-aerated soils remains limited⁹⁻¹¹. The smaller decomposition rates of native SOC for subsoil than topsoil under temperature and moisture-controlled conditions in the absence of roots indeed demonstrates that the slowing of SOC dynamics with depth cannot be merely driven by abiotic conditions such as cold or waterlogged pedoclimate and oxygen limitation at depth. Furthermore, our result of a decline in γ_{SOC} with depth contrasts with the idea that increasingly anaerobic conditions within microaggregates with depth lead to deep SOC persistence even for well-aerated soils. This is because it should theoretically involve an increase in γ_{SOC} by preferential preservation of highly reduced organic matter¹⁵. This mechanism may however be central in permanently waterlogged soils in wetlands and peatlands^{16,17}. ”*

Comment: Line 280: this is the first time this point has come up, and I think it needs to be dealt with in the intro or people will be wondering why (like I did).

Response: As explained above, a dedicated sentence about abiotic limitations on biological activity at depth has been included in the introduction following your suggestion.

Comment: Supplementary lines 200-206: I am concerned about the addition of fertiliser to only the planted soils would this not lead to accelerated priming effects? If so, the planted and unplanted treatments are not comparable.

Response: Fertilizer has been added only to planted subsoils. As mentioned in the previous version of the manuscript, it was critical to fertilize only the planted subsoil and the not unplanted subsoil to avoid excessive concentrations of mineral nutrients, which already tend to accumulate in soils in absence of rhizodeposition and nutrient uptake by plant roots¹⁸⁻²⁰. Could this fertilization of planted subsoils have led to accelerated rhizosphere priming effects? Most studies on this topic showed that fertilization have usually small and rather negative effects on both priming effects^{21,22} and rhizosphere priming effects^{23,24}. Given this literature pattern and the fact that our fertilization allowed to reach similar soil mineral N concentrations between planted topsoil and subsoil microcosms (Supplementary Table 8), we are confident that the larger rhizosphere priming effects for subsoil than topsoil is not driven by our fertilization treatment.

Comment: Lines 442: What happened to the 25-40cm deep soil? Does this not represent an interesting transition zone?

Response: We agree that it might have been interesting to include an additional soil layer modality with intermediate depth to test whether the depth effect is rather linear or nonlinear with a potential threshold level. However, we rather chose to keep our experiment simple by not including this intermediate depth since the test of our main hypothesis required many measurements that we could not afford to apply on more soil layers.

Comment: Line 445: this is rather hot, nitrogen-based molecules are likely to be inaccurate much above 70°C. Using 105°C comes with the caveat that organic matter is likely to be lost (Reynolds 1970, Journal of Hydrology, 11, 258-273; Lekshmi et al. 2014, Measurement, 54, 92-105).

Response: We are sorry about that but there has been a mistake in reporting this actually. In fact, we proceeded following standard methodology with soil been air-dried, while only a subset of soil been dried at 105 °C to measure soil moisture. This point has been clarified in the manuscript:

L. 396-397. *“These soil cores were first sieved at 2 mm and a portion was air-dried at room temperature. A portion of the air-dried soil was also ground (<250 μ m) to homogeneity. ”*

Reviewer #2 (Remarks to the Author):

Manuscript NCOMMS-22-26671 tests a novel bioenergetics-based framework for soil organic matter persistence as a function of soil depth. The dynamics and persistence of soil organic matter has received a great deal of attention over the past 20+ years. The community has undergone a paradigm shift in understanding the major controls on soil organic matter persistence - from chemical composition of recalcitrant moieties, to mineral association and ecosystem properties. These are well synthesized by Schmidt et al 2011, cited in the manuscript. The challenge from the new paradigm, however, has been how to make the proposed, emergent property of ecosystem-level persistence measurable, modelable, and predictable. And hence it appears that a new paradigm is being developed, based on bioenergetics and rooted in thermal analysis techniques. This manuscript represents an important contribution to this new paradigm development, with strong conceptual underpinnings and appropriate and rigorous experimental design and data.

The experimental study is well designed. All methods are appropriate, and calculations appear to be correct and valid. The combination of multiple thermal analyses methods, along with incubations and isotopic analyses make for a robust set of observations that support the conclusions drawn by the authors. When not necessarily creating an extensively comprehensive dataset (as would a meta-analysis of a large, global dataset), the inclusion of soils with differing mineralogy (ie, surface reactivity) makes the observations even more robust. Reporting and visualization of the results is highly appropriate, with some caveats:

Comment: In Table 1, "Depth effect" might be better reported as an actual statistical outcome. That is, rather than a mean relative/absolute change, it could be reported as the ANOVA effect size (which can be calculated multiple ways). Alternatively, a statistical test (p-value, etc.) might be reported to demonstrate the degree to which the difference falls outside the range attributable to natural variability.

Response: A new column giving the parameter 'partial η^2 ' has been added to provide to the ANOVA effect size about depth effect. We specified below Table 1 what represent partial η^2 :

L. 879-880. "*Partial η^2 indicates the proportion of variance associated with the depth effect after accounting for soil type effect.*"

We also fully explained how it was calculated in the supplementary methods:

L. 318-322. "*Partial η^2 of depth effect on SOC properties is calculated as the sum of squares for the depth effect divided by the total sum of squares (after accounting for the variance associated with soil type effect). It was computed using the 'eta_squared' function of the 'effectsize' package²⁵ on a linear mixed-effect model fitted using the 'lmer' function of the 'lme4' package²⁶ and including 'soil layer' as a fixed factor and 'soil type' as a random factor.*"

Comment: There are a large number of loading arrows in Fig 2a. Perhaps only show those that are "significant" in that they are deemed the most important and worth of explanation in the main body of the text.

Response: To help our readers catching the important information in the ordination without being overwhelmed by too much data, we reduced the number of variables from 25 to 18 in Fig 2. We basically removed the variables that are somewhat redundant or only corroborating other variables. For instance, the three variables about thermal stability has been removed since they only confirm the trend about activation energy. We similarly removed the HI and OI indices, which mainly confirmed the trend about the degree of reduction of SOC. The variable fMOAM has been removed because it is strictly equal to 1-fPOM. The variable Si_o and Al_o has been merged (Si_o+Al_o) since they both inform us about aluminosilicates. Note that we maintained the presence in Table 1 of the variables removed in the ordination since they remained interesting for the readers.

Comment: The caption refers to the width of the decomposition arrow being proportional, but the differences in width are perhaps too subtle to be immediately noticeable. Perhaps elements of the cartoon can be further exaggerated to highlight the points being made.

Response: *This is a good idea. We modified Fig. 6 according to your suggestion. We took that opportunity to further improve this figure by showing the depth patterns in SOC persistence, accessibility, energy quality, energy limitation of SOC decomposition and root density and energy supply.*

The manuscript is very well written and I have very few, very minor editorial comments (see below).

Comment: However, I did find that a large portion of the discussion section (lines 245 through 343) is largely an elaboration of what is in the results section. Perhaps it's because the results section is written sufficiently clearly and understandably, the first part of the discussion seems somewhat redundant. Going forward from line 344 is what I consider to be a discussion in the conventional sense. Perhaps the discussion section as a whole can be shortened for conciseness. Alternatively, the goals of the elaboration of the results can be to place observations in context with other values reported in the literature rather than a repetition of what is in the results section. I understand there are likely few reported data to compare to, as this is an evolving paradigm with novel experiments. Another possibility would be to elaborate on the conceptual framework (both historically as it seems related to the early ecosystem work of Odum) and then place the observations within this framework, rather than to elaborate on the results themselves.

Response: *We agree that there was quite a lot of redundancy between the results section and the beginning of the discussion section, probably because we were too focused on discussing of all the results point-by-point and compared them to previous studies. Following your first suggestion, we shortened the discussion section for conciseness by reformatting it into a more synthesis version centered around our hypothesis and key findings (L. 254 to 370). This allowed to reduce the length of the discussion section by 20 %, that is from 1,844 to 1,516 words.*

Comment: In terms of editorial comments, the text is largely free of grammatical and linguistic errors. However, I would recommend that the terms "higher" and "lower" be reserved for vertical position since the study involves samples taken higher and lower in the soil profile. Instead, these terms should be replaced in comparisons with terms that are more directly equivalent to ">" and/or "<", such as "larger/smaller", "greater/lesser", etc. Also...

ln46: "reactivity contexts" instead of "context"

ln60: "30 cm deep" instead of "depth"

ln101: "To date" instead of "So far"

ln106 and throughout: The term "field" implies measurement potential performed "in-situ" and literally in the field. I think this term can be omitted in most usages in the manuscript.

ln109: I don't think "genericity" is a word, and even if so, it's awkward. Replace with "robustness"

ln128: "Microbial biomass" instead of "biomasses"

ln162: "in soil columns" instead of "on soil columns"

ln167: "consisted of" instead of "in"

ln218: "high amounts" instead of "amount"

ln219: "mineral composition" instead of "nature"

ln219: omit "indeed"

ln226: omit "in contrast"

ln351: "have often" instead of "has often"

Response: *We performed all the corrections you recommended. Thanks for your careful review that helped us improve the manuscript.*

Overall, I find the manuscript to be of sufficient rigor, quality and novelty for publication in Nature Communications with only minor revisions.

Response: *We warmly thanks you for your consideration of our work.*

Prof. Alain Plante

University of Pennsylvania

References

1. Eilers, K. G., Debenport, S., Anderson, S. & Fierer, N. Digging deeper to find unique microbial communities: The strong effect of depth on the structure of bacterial and archaeal communities in soil. *Soil Biology and Biochemistry* **50**, 58–65 (2012).
2. Fierer, N., Schimel, J. P. & Holden, P. A. Variations in microbial community composition through two soil depth profiles. *Soil Biol. Biochem.* **35**, 167–176 (2003).
3. Naylor, D., McClure, R. & Jansson, J. Trends in Microbial Community Composition and Function by Soil Depth. *Microorganisms* **10**, 540 (2022).
4. Turner, S. *et al.* Microbial Community Dynamics in Soil Depth Profiles Over 120,000 Years of Ecosystem Development. *Frontiers in Microbiology* **8**, (2017).
5. Jobbágy, E. G. & Jackson, R. B. The vertical distribution of soil organic carbon and its relation to climate and vegetation. *Ecol. Appl.* **10**, 423–436 (2000).
6. Hiederer, R. & Köchy, M. *Global soil organic carbon estimates and the harmonized world soil database.* (2011).
7. Schmidt, M. W. I. *et al.* Persistence of soil organic matter as an ecosystem property. *Nature* **478**, 49–56 (2011).
8. Dungait, J. A. J., Hopkins, D. W., Gregory, A. S. & Whitmore, A. P. Soil organic matter turnover is governed by accessibility not recalcitrance. *Glob. Change Biol.* **18**, 1781–1796 (2012).
9. Mathieu, J. A., Hatté, C., Balesdent, J. & Parent, É. Deep soil carbon dynamics are driven more by soil type than by climate: a worldwide meta-analysis of radiocarbon profiles. *Glob. Change Biol.* **21**, 4278–4292 (2015).
10. Chen, L. *et al.* Soil carbon persistence governed by plant input and mineral protection at regional and global scales. *Ecology Letters* **24**, 1018–1028 (2021).
11. Salomé, C., Nunan, N., Pouteau, V., Lerch, T. Z. & Chenu, C. Carbon dynamics in topsoil and in subsoil may be controlled by different regulatory mechanisms. *Glob. Change Biol.* **16**, 416–426 (2010).
12. Roth, V.-N. *et al.* Persistence of dissolved organic matter explained by molecular changes during its passage through soil. *Nature Geoscience* (2019) doi:10.1038/s41561-019-0417-4.
13. Schrumpf, M. *et al.* Storage and stability of organic carbon in soils as related to depth, occlusion within aggregates, and attachment to minerals. *Biogeosciences* **10**, 1675–1691 (2013).
14. Rumpel, C. & Kögel-Knabner, I. Deep soil organic matter—a key but poorly understood component of terrestrial C cycle. *Plant and Soil* **338**, 143–158 (2011).
15. Keiluweit, M., Wanzek, T., Kleber, M., Nico, P. & Fendorf, S. Anaerobic microsites have an unaccounted role in soil carbon stabilization. *Nature Communications* **8**, 1771 (2017).
16. Boye, K. *et al.* Thermodynamically controlled preservation of organic carbon in floodplains. *Nature Geoscience* **10**, 415–419 (2017).
17. LaRowe, D. E. & Van Cappellen, P. Degradation of natural organic matter: A thermodynamic analysis. *Geochimica et Cosmochimica Acta* **75**, 2030–2042 (2011).
18. Zhang, Q. *et al.* A distinct sensitivity to the priming effect between labile and stable soil organic carbon. *New Phytologist* **n/a**, (2022).

19. Henneron, L., Kardol, P., Wardle, D. A., Cros, C. & Fontaine, S. Rhizosphere control of soil nitrogen cycling: a key component of plant economic strategies. *New Phytol.* **228**, 1269–1282 (2020).
20. Dijkstra, F. A., Bader, N. E., Johnson, D. W. & Cheng, W. Does accelerated soil organic matter decomposition in the presence of plants increase plant N availability? *Soil Biology and Biochemistry* **41**, 1080–1087 (2009).
21. Feng, J., Tang, M. & Zhu, B. Soil priming effect and its responses to nutrient addition along a tropical forest elevation gradient. *Glob. Change Biol.* **27**, 2793–2806 (2021).
22. Feng, J. & Zhu, B. Global patterns and associated drivers of priming effect in response to nutrient addition. *Soil Biology and Biochemistry* **153**, 108118 (2021).
23. Jiang, Z., Liu, Y., Yang, J., Zhou, Z. & Gunina, A. Effects of nitrogen fertilization on the rhizosphere priming. *Plant and Soil* (2021) doi:10.1007/s11104-021-04872-6.
24. Cheng, W., Johnson, D. W. & Fu, S. Rhizosphere Effects on Decomposition: Controls of Plant Species, Phenology, and Fertilization. *Soil Sci. Soc. Am. J.* **67**, 1418–1427 (2003).
25. Ben-Shachar, M. S., Lüdtke, D. & Makowski, D. Effectsize: estimation of effect size indices and standardized parameters. *Journal of Open Source Software* **5**, 2815 (2020).
26. Bates, D., Mächler, M., Bolker, B. & Walker, S. Fitting Linear Mixed-Effects Models Using lme4. *Journal of Statistical Software; Vol 1, Issue 1 (2015)* (2015) doi:10.18637/jss.v067.i01.